# A multiplexed bacterial two-hybrid for rapid characterization of protein–protein interactions and iterative protein design

W. Clifford Boldridge [1,14], Ajasja Ljubetič[2,3,14] ✉, Hwangbeom Kim[1,11], Nathan Lubock[1,12], Dániel Szilágyi [4], Jonathan Lee [5,13], Andrej Brodnik[4], Roman Jerala [2,3] ✉ & Sriram Kosuri[1,12,6,7,8,9,10] ✉

Protein-protein interactions (PPIs) are crucial for biological functions and have applications ranging from drug design to synthetic cell circuits. Coiled-coils have been used as a model to study the sequence determinants of specificity. However, building well-behaved sets of orthogonal pairs of coiled-coils remains challenging due to inaccurate predictions of orthogonality and difficulties in testing at scale. To address this, we develop the next-generation bacterial two-hybrid (NGB2H) method, which allows for the rapid exploration of interactions of programmed protein libraries in a quantitative and scalable way using next-generation sequencing readout. We design, build, and test large sets of orthogonal synthetic coiled-coils, assayed over 8,000 PPIs, and used the dataset to train a more accurate coiled-coil scoring algorithm (iCipa). After characterizing nearly 18,000 new PPIs, we identify to the best of our knowledge the largest set of orthogonal coiled-coils to date, with fifteen on-target interactions. Our approach provides a powerful tool for the design of orthogonal PPIs.

Protein–protein interactions (PPIs) are integral to most biological functions and are required for such diverse processes as cell division, signalling, metabolism, transcription and translation[1]. Our ability to design and create functions and structures as complex as those found in nature, though still in its infancy, is progressing with advances in both protein design algorithms and gene synthesis.

For example, designed orthogonal sets of interacting proteins, in which each pair of proteins interacts only with its intended on-target pair and none of the other, off-target proteins present in the set, can be used to build nanoscale superstructures for applications in biology, biological engineering and materials science[2]. Supramolecular protein designs can be created using simple, natural protein families such as coiled-coils, which have been used to build numerous designed protein assemblies[3–5]. However, identifying orthogonal natural proteins is difficult because evolutionarily related proteins often display significant cross-interactions. Another method is to computationally

[1]Department of Chemistry and Biochemistry, University of California, Los Angeles, CA 90095, USA. [2]Department of Synthetic Biology and Immunology, National Institute of Chemistry, 1000 Ljubljana, Slovenia. [3]EN-FIST Centre of Excellence, 1000 Ljubljana, Slovenia. [4]University of Primorska, 6000 Koper, Slovenia. [5]Department of Chemical and Biomolecular Engineering, University of California, Los Angeles, CA 90095, USA. [6]UCLA-DOE Institute for Genomics and Proteomics, University of California, Los Angeles, Los Angeles, CA 90095, USA. [7]Molecular Biology Institute, University of California, Los Angeles, Los Angeles, CA 90095, USA. [8]Institute for Quantitative and Computational Biosciences, University of California, Los Angeles, Los Angeles, CA 90095, USA. [9]Eli and Edythe Broad Center of Regenerative Medicine and Stem Cell Research, University of California, Los Angeles, Los Angeles, CA 90095, USA. [10]Jonsson Comprehensive Cancer Center, University of California, Los Angeles, Los Angeles, CA 90095, USA. [11]Present address: Samsung Biologics, Incheon, Republic of Korea. [12]Present address: Octant Inc, Emeryville, CA 94608, USA. [13]Present address: Keck School of Medicine, University of Southern California, Los Angeles, CA 90033, USA. [14]These authors contributed equally: W. Clifford Boldridge, Ajasja Ljubetič. ✉e-mail: ajasja.ljubetic@ki.si; roman.jerala@ki.si; skosuri@chem.ucla.edu

design de novo proteins; in particular, Rosetta-based designs have produced homodimers[6,7] and heterodimers[8]. In a state-of-the-art example, Chen et al.[8] performed a single-pot experiment mixing fifteen designed heterodimer pairs, which resulted in a set of twelve orthogonal heterodimers. However, the Rosetta energy function did not successfully predict orthogonality, and predicting orthogonal binding and designing large orthogonal sets remain beyond current de novo design methods[3].

Coiled-coils, in particular, have many useful characteristics in terms of creating atomically precise designs for macromolecular structures. They are small and precisely oriented, and numerous sequence-based and parametric models exist with which to describe their properties. First identified at the dawn of molecular biology by both Pauling[9] and Crick[10], coiled-coils are defined by their heptad repeat HPPHPPP (H = hydrophobic residue, P = polar residue), represented as positions **abcdefg**. This relatively simple structure has given rise to many computational models describing coiled-coil interactions, from the parametric Crick equations of 1953[11] to contemporary linear models[12-15]. However, because of their shared similar structure, building large sets of orthogonally interacting coiled-coils, in which all on-target interactions are favoured over all off-target interactions, remains difficult. Though numerous groups have attempted to create orthogonal sets of coiled-coils, these sets have been limited in size and displayed significant off-target interactions[15,16]. Increasing our ability to build and characterise large sets of interacting proteins could help solve this problem by providing empirical data with which to improve computational models of PPIs. Simultaneously, this would vastly increase the number of available orthogonal building blocks for nanoscale structural design, allowing for the creation of previously unbuildable structures.

Here, we combine gene synthesis, an assay that allows for multiplexed bimolecular interaction screening, and a computational pipeline to design large libraries of orthogonally interacting coiled-coils. We first built and validated the next-generation bacterial two-hybrid (NGB2H) system, which has several unique advantages over other methodologies in terms of characterising protein libraries. In particular, the NGB2H system allows for the screening of bimolecular interactions without having to test all-against-all libraries; direct large-scale synthesis using oligonucleotide arrays to explore the design space; quantitative readouts on an entire library, including negative interactions, and the characterisation of low-affinity interactions inside the crowded cellular context. We did this iteratively, with synthetically designed libraries increasing in size from 256 interactions to more than 18,000 interactions. From this, we identified to the best of our knowledge the largest sets of orthogonal proteins to date and developed an improved coiled-coil scoring algorithm (iCipa) for use in future investigations of this versatile protein domain.

## Results
### NGB2H system design
Despite a wealth of techniques via which to analyse PPIs, there is not currently a method that facilitates high-throughput characterisation when analysing PPIs in formats other than all-against-all or is able to distinguish between closely related constructs. However, such a system would allow investigations of PPIs within protein families, polymorphic PPIs, and de novo designed PPIs that are currently intractable. Thus, we built a generalisable, scalable bacterial two-hybrid system using a significantly modified version of the *B. pertussis* adenylate cyclase two-hybrid[17] (Fig. 1A, Supplementary Information Section 1). Briefly, the two-hybrid functions much as in Karimova et al.[17], in which interacting hybrid proteins reconstitute adenylate cyclase to produce cAMP, which drives reporter gene expression. We measured the relative transcription of a uniquely identifying DNA barcode residing in the reporter gene, which serves as a measure of interaction strength. The barcode is mapped to the two fully sequenced hybrid proteins at an

early cloning step using high-throughput sequencing when the barcode and proteins are physically adjacent. This unambiguously identifies even highly homologous proteins and separates synthetic errors from programmed designs. Thus, measuring the relative barcode transcription provides a quantitative, massively multiplexed characterization of PPIs with short-read sequencing. Because the NGB2H system uses a mapping step, it can use gene synthesis, rather than preconstructed libraries, to create diversity, which further frees it from the one-against-all or all-against-all testing common in two-hybrids. We made a number of other improvements, including (1) titratable and inducible control of hybrid protein expression and optimised reporter response on a single plasmid, (2) a background strain with linear cAMP accumulation, (3) a green fluorescence protein (GFP) reporter instead of beta-galactosidase for more rapid individual characterisation, (4) the use of multiple barcodes per construct to achieve statistically robust results and (5) a scarless cloning scheme that allows for library creation with any designed sequence (more information in Supplementary Information Section 1).

### Validation of the NGB2H system
After optimising the system with single-construct GFP measurements (Supplementary Fig. 1), we validated the NGB2H system with 256 previously characterised interactions[15], which we call the CC0 Library. The CC0 Library is a set of sixteen de novo designed, orthogonal, heterodimeric coiled-coils that are tested in an all-against-all configuration. The proteins are highly similar, being four heptad coiled-coils that vary only at the **a**-position (Ile/Asn), **e**-position and **g**-position (Lys/Glu) (Fig. 1B). We designed the CC0 Library to be compatible with our system (Supplementary Fig. 2A) and then barcoded and cloned it (Supplementary Figs. 3A, 4). After inducing the two-hybrid for six hours, we took samples for RNA and DNA extraction to measure the interaction strength and normalize for plasmid abundance, respectively. We obtained high-quality measurements for all 256 protein pairs and calculated an interaction score, defined as the natural logarithm of the median of the ratio of the RNA to DNA reads: $Interaction\ score = \ln\left(median\left(\frac{RNA\ reads}{DNA\ reads}\right)\right)$.

Only barcodes for which ten or more reads were obtained in every DNA replicate and that perfectly mapped to designed protein pairs were used in further analysis. The NGB2H assay was highly replicable, with biological replicates having similar interaction scores (Pearson's $r > 0.98$, $p < 10^{-15}$), with a dynamic range of more than 100-fold (Fig. 1C).

We checked several internal controls to validate the measurements of the NGB2H assay. First, because the protein code is degenerate, we screened nine codon usages for each pair of proteins. Different codon usages showed consistent interaction scores (representative pair Fig. 1D), with all usages correlating with Pearson's $r > 0.92$ and $p < 10^{-15}$ (Supplementary Fig. 5), demonstrating minimal effects on the part of DNA sequence variation and low levels of noise in the interaction scores. We also compared the interaction scores of protein pairs when the two constituent proteins were attached to the other half of the two-hybrid, which we call the reciprocal orientation. We found that the CC0 Library exhibits a strong correlation between the primary and reciprocal orientations (Pearson's $r = 0.92$, $p < 10^{-15}$, Fig. 1E), indicating that the biological machinery of the NGB2H system faithfully recapitulates the biochemical interaction. In addition, a portion of our library contained frameshift mutations, which should not create functional PPIs. As expected, the interaction scores of constructs with indels are clustered at the bottom of the range of correct constructs (Supplementary Fig. 6). Last, to show that the NGB2H system does not suffer from barcode effects or selection pressure from the repeated cloning steps, we replicated the assay with an independent re-barcoding and re-cloning of the CC0 Library, which showed strong correlation with the first iteration's interaction scores (Pearson's $r > 0.98$, $p < 10^{-15}$, Fig. 1F).

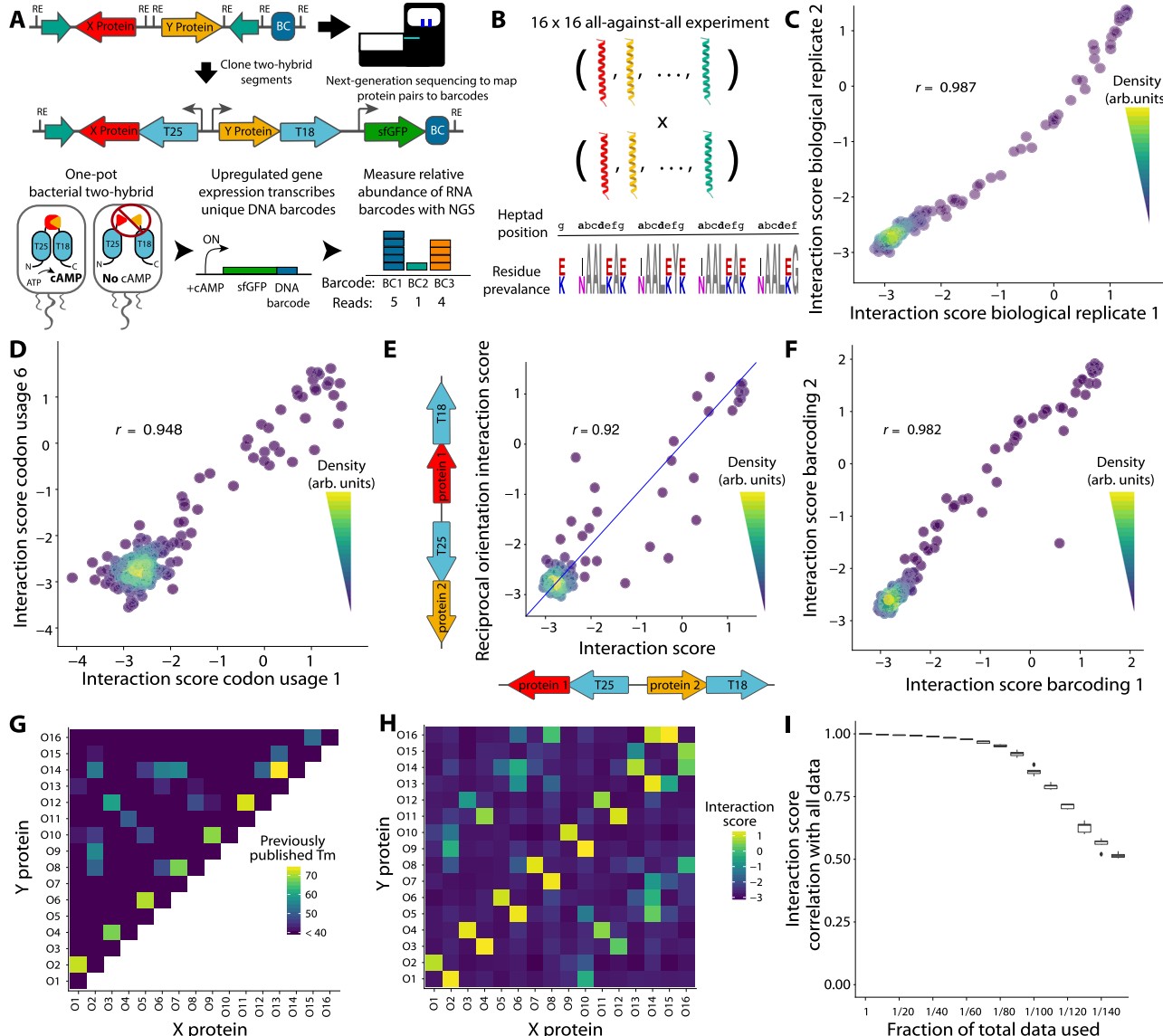

**Fig. 1 | Design and validation of the NGB2H assay. A** Top: schematic of the next generation bacterial two-hybrid (NGB2H) system cloning and construct. T25, T18 - adenylate cyclase halves; BC - unique DNA barcode identifying the protein pair. Bottom: Workflow of NGB2H system. Interacting proteins reconstitute adenylate cyclase, producing cAMP (cyclic adenosine monophosphate), which drives the gene expression of the barcoded super-folder green fluorescent protein (sfGFP) reporter. Relative barcode abundance is quantified using next-generation sequencing (NGS). **B** The CC0 Library is composed of 16 coiled-coils tested against one another. (Bottom) Sequence logo representing the diversity in the CC0 Library. Residues that vary are shown in colour. **C** Interaction scores of CC0 library members are consistent between biological replicates (Pearson's r > 0.98). **D** Two different codon usages have consistent interaction scores (Pearson's r > 0.94, representative sample). **E** Interaction strength is similar (Pearson's r > 0.92) regardless of which protein is attached to which half of adenylate cyclase. The blue line represents y = x. **F** Interaction scores of separately barcoded, cloned, and tested replicates are consistent (Pearson's r > 0.98). **G** Published circular dichroism (CD) melting point (Tm) data. **H** Experimentally determined interaction scores. **I** CC0 library Raw data can be subsampled and still correlate well with the full dataset. Boxplot center lines represent the median, the hinges represent the 25th and 75th percentiles and whiskers represent the largest/smallest value within 1.5x it's respective hinge for 50 subsamples with replacement of the full data. Source data are provided as a Source Data file.

Having confirmed the internal consistency of the CC0 Library, we compared it to the previously published results. Compared to the circular dichroism data published in Crooks et al.[15], we found that the NGB2H system's dynamic range correlated well with melting temperatures greater than 40 °C (Fig. 1G, H). Given the differences in technique – in vivo versus in vitro, interaction strength versus helicity – the correlation between the interaction score and melting point temperatures (Pearson's $r > 0.75$, $p < 10^{-15}$, Supplementary Fig. 7) largely validate the NGB2H system. Finally, the NGB2H system must be highly scalable. To test its scalability, we computationally reduced the number of reads used in the analysis between 10 and 150-fold and found strong agreement with our full dataset, even when the raw data

were reduced 100-fold (Pearson's $r > 0.85$, $p < 10^{-15}$, Fig. 1I), which implies the ability to accurately screen ~25,000 interactions at a similar read depth.

## Design of large sets of orthogonal coiled-coils

All dimeric coiled-coils have a similar structure, which is why sequence-based scoring functions can fruitfully predict melting temperatures or binding affinities. The scoring functions accept two sequences as input, usually beginning with a specific register, and return a score. One of the widely used algorithms is bCipa[14], which is based on summing weights for residue-residue interaction pairs, as well as electrostatic interactions and helical propensity, and predicts melting

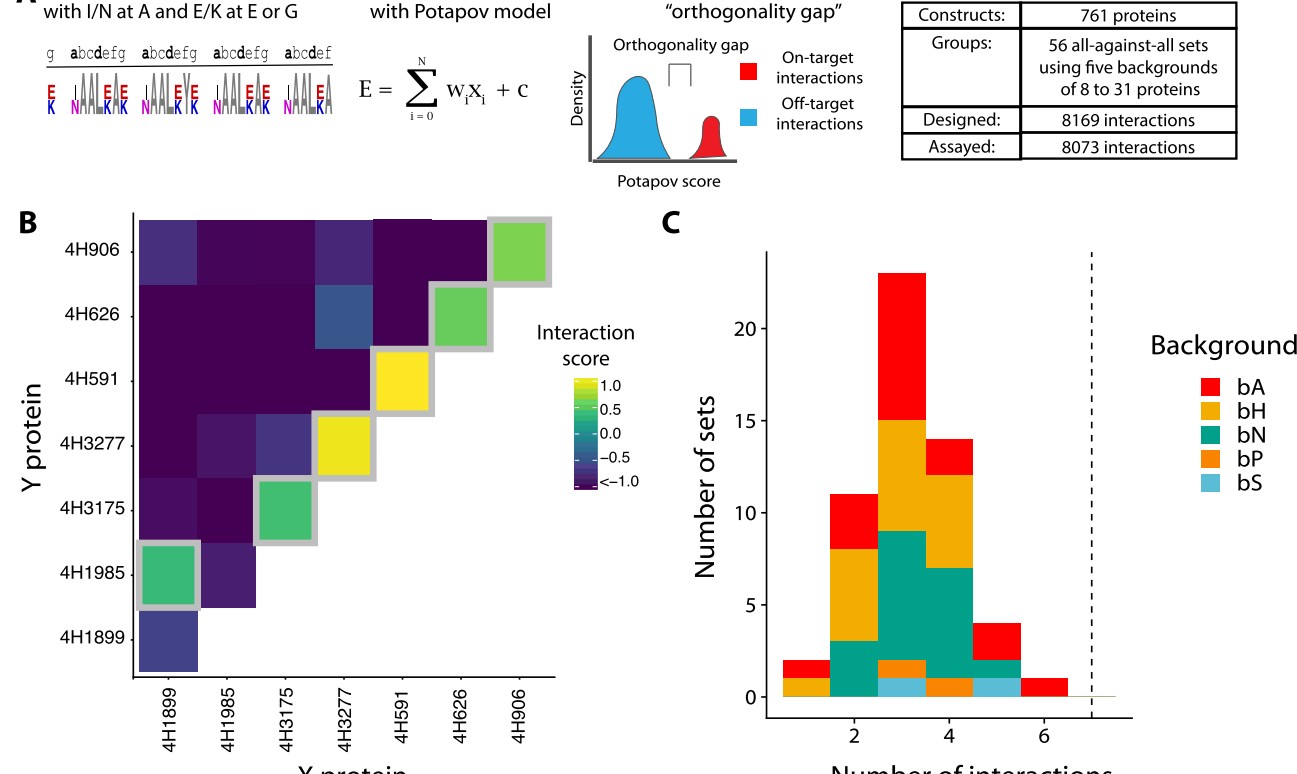

**Fig. 2 | Large orthogonal subsets of coiled-coils from the CCNG1 library.**
**A** Schematic of the CCNG1 Library design. All four-heptad coiled-coils with variation at the **a**-, **e**-, and **g**- positions were scored for interactions with the model of Potapov et al., and subsets of coiled-coils with large orthogonality gaps were identified. In total, we designed and tested 56 sets of orthogonal coiled-coils. **B** The orthogonal subset of coiled-coils with the largest number of on-target interactions (six on-target interactions). Grey boxes identify on-target interactions. **C** Number of interactions per orthogonal subset of coiled-coils. Dashed line represents the number of on-target orthogonal interactions in the CC0 library. Colours show the different backgrounds used, while the interfacial residues remained the same. Source data are provided as a Source Data file.

temperatures. The state-of-the-art scoring function was developed by Potapov et al.[13], which uses triplet weights, in addition to the pair weights, and a much larger training set to predict the free energy of binding. The paper also benchmarks the most common CC scoring functions, such as Fong/SVM[18] and Vinson/CE[12].

To computationally predict large, orthogonal sets of coiled-coils for empirical verification, we built a two-step computational pipeline (Fig. 2A). In brief, we calculated 16.7 million scores for all dimeric interactions between four-heptad coiled-coils with Ile or Asn at the **a**-position and Glu or Lys at the **e**- and **g**- positions using the scoring model of Potapov et al.[13]. The surface **b**-, **c**- and **f**-positions were set to Ala. We then identified orthogonal sets, which can be divided into *on-target* and *off-target* interactions such that each constituent protein participates in exactly one on-target interaction, which is stronger than every off-target interaction. This allows us to define an *orthogonality gap* for an orthogonal set, where the orthogonality gap is calculated as the weakest on-target interaction minus the strongest off-target interaction. For example, in Fig. 2B, on-target interactions are on the diagonal (homodimers) or just above the diagonal (heterodimers). All other interactions are considered off-target. Though computationally challenging, identifying sets with an orthogonality gap is tractable as a variant of the maximum independent set problem[19]. Using the bCipa and Potapov scoring functions, we identified the fifteen largest sets and included each of them with three different sets of residues at the **b**-, **c**- and **f**-positions because surface positions can modulate dimer stability and solubility[20]. We refer to a set of residues used at the **b**-, **c**- and **f**- positions as *backgrounds* because these do not affect orthogonality. We combined these with

two sets of controls spanning eleven backgrounds, resulting in a total of 56 sets containing between 64 to 961 interactions (8169 interactions overall), which we named the CCNG1 Library. After testing a subset of the CCNG1 Library to validate our in-house designs, which we call the CC1 Library, (see Supplementary Figs. 8, 9; Supplementary Information Section 8.3), we designed (Supplementary Fig. 2C), cloned (Supplementary Fig. 3C, 4), and performed the NGB2H assay, from which we collected quality data (Supplementary Fig. 10) on 8073 interactions. The CC0 Library was added to the CCNG1 library as an internal control (Supplementary Fig. 10C).

The space of all possible pairs, assuming only our limited set of amino acid residues (~16 M), is several orders of magnitude larger than what could be screened experimentally (~25 k), so the design process is crucial in identifying feasible orthogonal sets that can be experimentally tested.

## Large orthogonal sets in the CCNG1 library
Although we designed our coiled-coils to form orthogonal sets, the current state-of-the-art coiled coil scoring functions are not sufficiently accurate to do so reliably, and nearly all sets contained off-target interactions stronger than some of the on-target pairs. The proteins involved in strong off-target interactions can be removed from the set, leaving only those interactions that are experimentally verified to be orthogonal. Thus, we refer to an *orthogonal subset* as the largest experimentally characterised group of orthogonal interactions among what was computationally predicted to be an orthogonal set. To identify the orthogonal subset of each designed orthogonal set, we used a similar approach to that described above and reduced the

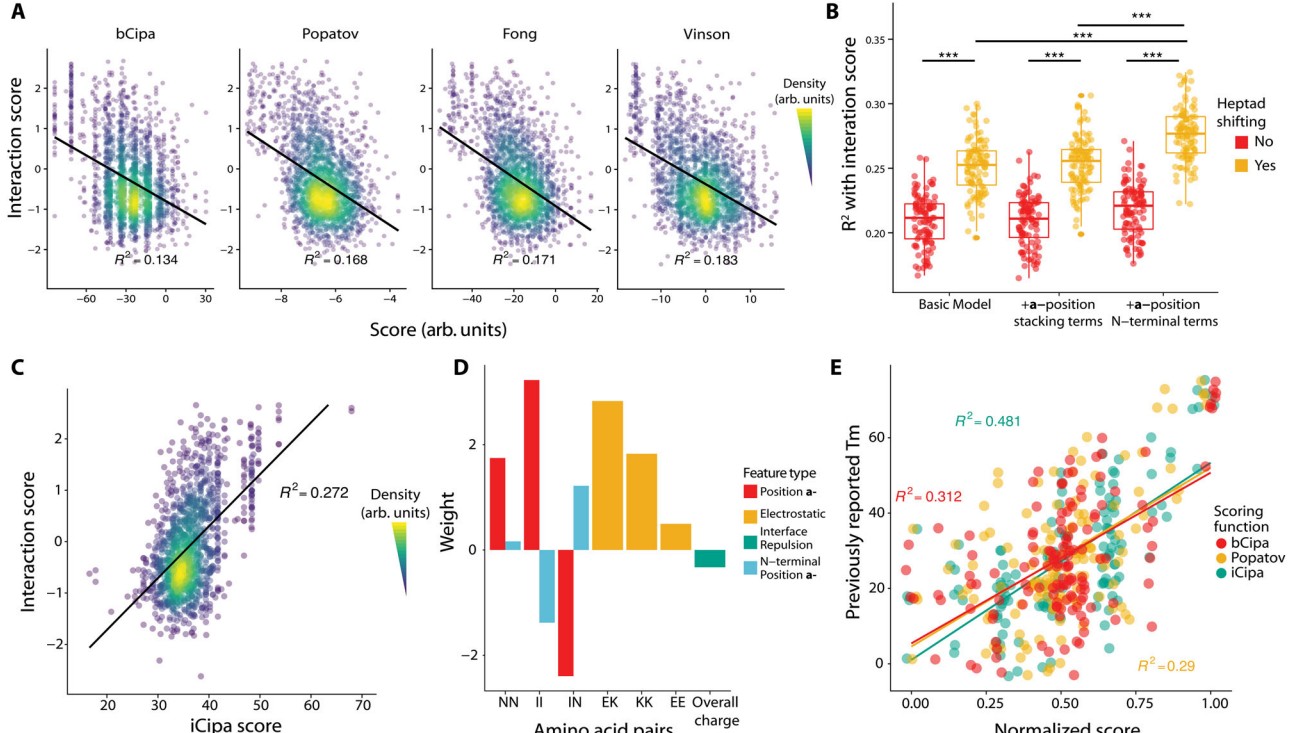

**Fig. 3 | Comparison, development and validation of the iCipa model. A** Previous models of coiled-coils predict the interactions in the CCNG1 with low $R^2$. The black line represents a linear model of the interaction scores predicted by different algorithms. **B** Coefficient of determination of interaction scores of different iCipa candidates evaluated during development. Each point represents one bootstrap of the data. $N = 100$ bootstraps. Boxplot center lines represent the median, the hinges represent the 25th and 75th percentiles and whiskers represent the largest/smallest value within 1.5x it's respective hinge. \*\*\*$p < 10^{-15}$ by two-tailed $t$-test. **C** iCipa is more predictive of interaction scores ($R^2 > 0.27$) than the previous models shown in (**A**). Black line represents a linear model of interactions scores, as predicted by iCipa

scores. **D** Weights for the iCipa model. Each weight scores a pair of amino acid residues at specific registers between the coiled coils (at **a-a'**: NN and II, at **e-g'** and **g-e**: KE, KK and EE). **E** iCipa is more predictive of the previously published CC0 melting points than the bCipa or Potapov scoring functions. Individual dots represent melting points as compared with the normalised score from one of the three scoring algorithms. Boxplot center lines represent the median, the hinges represent the 25th and 75th percentiles and whiskers represent the largest/smallest value within 1.5x it's respective hinge. Source data are provided as a Source Data file.

problem to the maximum independent set problem using Interaction scores from the NGB2H assay.

To make our results robust to experimental noise from the NGB2H assay, we needed to find an appropriate orthogonality gap, that is larger than the uncertainty of the interactions score. We have performed a thorough analysis of both the CC0 internal control (technical repeats), external controls (comparison to measured melting points, Supplementary Fig. 10C) and especially the availability of reciprocal enzyme orientations for the same peptide pair (pairs of identical peptides where the split cAMP parts are reversed). We found uncertainty of less than 0.8 interaction scores in all experiments in this paper (Supplementary Data 9). Thus, to be conservative we enforced an orthogonality gap of at least 1.0 Interaction Score. Using this framework, we were able to identify an orthogonal subset of coiled-coils that contains six pairs, which includes one heterodimer and five homodimers (Fig. 2B). The orthogonality gap we enforce is very strict, for example the CC0 control set has a gap of only 0.4, and at the orthogonality gap of 1.0 it contains only four pairs instead of seven.

There are also applications where the requirements for orthogonality can be reduced, for example in building protein origami as demonstrated by Aupič et al.[21], in which two identical pairs were used in the same structure. Pairwise orthogonality is the most stringent criterion. In a single-pot experiment, in which all pairs would be present, we speculate that orthogonality would only improve because the off-target states would be competing with the on-target states.

Therefore we have also calculated orthogonal sets with orthogonality gaps of 0.0 and 0.5. At an orthogonality gap of zero, 20 of our

51 experimentally identified orthogonal subsets in CCNG1 library had more than the seven on-target orthogonal interactions (Supplementary Fig. 13). Orthogonal sets at different orthogonality gaps are presented in Supplementary Data 6.

The CCNG1 Library represents the first large-scale systematic investigation of the effects of variation at the **b**-, **c**-, and **f**-positions; therefore, we sought to understand how these positions influenced interactions. As expected, we found that different backgrounds did not significantly affect orthogonality (Fig. 2C and Supplementary Figs. 11, 12). We tested six backgrounds containing the same interfacial residues as the CC0 Library (Supplementary Fig. 14 and Supplementary Information Section 8.4) and found that charged but less helical backgrounds led to weaker, less specific interaction profiles. The findings agree with the model presented by Drobnak et al.[22], in which the **b**-, **c**- and **f**- positions were used to modulate affinity.

### Improvement of coil-coiled interaction-prediction algorithms
The CCNG1 Library dataset represents the largest dataset of coiled-coil interactions to date. We reasoned that our data could serve as a training set to improve on currently available models. To benchmark current models, we computed scores using the algorithms bCipa[14], Potapov/SVR[13], Fong/SVM[18] and Vinson/CE[12], which are all linear models with features for amino acid pairings. Each algorithm is only weakly predictive of our measured interactions with the bA background (Fig. 3A) because all models have an $R^2 < 0.2$. Notably, each algorithm predicted the strongest interactions well but also predicted many weak interactions that, when measured, had high interaction scores.

We built several linear models similar to bCipa, which included numerous innovations (Supplementary Information Section 3). First, we trained a model on our data that only included weights for the **a**-, **d**-, **e**- and **g**- position combinations. We also created versions of this simple model with terms for either consecutive residues in the **a**- position of the same protein or separate terms for weights at the N-terminal **a**- position, where fraying may occur (Supplementary Fig. 15A).

We then expanded these models with a scoring technique, which we call heptad shifts (Supplementary Fig. 15B). In short, we expect the predominant form of coiled-coil interaction to be the alignment of heptads that have the strongest interaction. In terms of the large number of off-target interactions, this does not necessarily indicate that all four heptads are aligned with the N-terminus but, rather, could indicate an interface of three or fewer heptads. We have trained the models iteratively by changing the alignment of off-target pairs, retraining the models and rescoring the off-target alignments until convergence was achieved (in less than five repetitions in all cases). All of our heptad-shifting scoring algorithms were significantly better than the corresponding non-shifting versions. Our N-terminal **a**- position weights algorithm was significantly better than both the basic algorithm and the consecutive **a**- position algorithm (Fig. 3B). Thus, our final model, which we call iCipa, uses heptad shifting and terms for the N-terminal **a**- positions, and it is more predictive of CCNG1 Interaction scores than previous models, with an $R^2 = 0.27$ (Fig. 3C). The effect of heptad shifting on iCipa, as well as bCipa and the Potapov scoring function, is shown in Supplementary Fig. 16.

iCipa is a linear model, which facilitates interpretation. The weights of iCipa have expected and unexpected characteristics (Fig. 3D). **a**- position residues prefer Ile/Ile pairings, tolerate Asn/Asn pairings between proteins and disfavour Ile/Asn pairings, as expected. As expected, the **e**- and **g**- positions favour salt bridges between Glu/Lys and disfavour Glu/Glu pairings. Perhaps counterintuitively, Lys/Lys pairings are acceptable, and previous biochemical work has identified mildly favourable binding contributions on the part of Lys/Lys pairings[23].

To test the iCipa model, we excluded all the data from the original CC0 Library while we trained the weights. When the scoring functions are normalised and compared (Fig. 3E), both the Potapov/SVR and bCipa algorithms performed worse in terms of predicting the measured melting temperatures, with $R^2 < 0.32$, as compared to iCipa, with $R^2 = 0.48$, representing a 50% increase in predictive ability. Importantly, the increase in predictive power for iCipa on the CC0 Library demonstrates that iCipa has not been trained on an artifact of the NGB2H system but, rather, that the NGB2H system provides high-quality data on PPIs, which can provide general insights into coiled-coil function.

## CCmax library design and verification

To evaluate iCipa's prediction capabilities, demonstrate the scalability of the NGB2H system, and identify larger orthogonal sets of coiled-coils, we built another library, the CCmax Library. The CCmax Library contains 18,491 interactions and contains 931 different coiled-coils in fifteen predicted orthogonal sets and seven control sets (Fig. 4A). The orthogonal sets were designed using our computational framework and scored with one of fifteen variants of iCipa. After designing (Supplementary Fig. 2D) and cloning, we collected high-quality data on 17,983 interactions (Supplementary Fig. 17). The CC0 Library was an internal control added to the CCmax Library, and it broadly agreed with its performance in our previous libraries (Supplementary Fig. 18).

## Orthogonal sets of the CCmax library

Similarly to the CCNG1 library, we identified the largest experimentally identified orthogonal subsets of each designed set with an orthogonality gap of 1.0 Interaction Score. These orthogonal subsets have as many as fifteen on-target pairs (Fig. 4B) and 318 total interactions from 18 different proteins (Supplementary Fig. 19). Five of the orthogonal

subsets contained more on-target interactions than the largest published coiled coil set[15]. Our largest orthogonal subset (Fig. 4C) contained fifteen coiled-coil dimers, twelve homodimers and three heterodimers, which is nine more on-target interactions than the set from CCNG1, showing the improvement of iCipa over bCipa and the Potapov scoring functions.

Similar to the CCNG1 Library, we also identified sets with lower orthogonality gaps of at least 0.0 Interaction Score, 0.5 Interaction Score, and one RMSD between the reported melting temperatures of the CC0 subset of the CCmax library mapped to Interaction Scores (Supplementary Fig. 17C). Lowering the orthogonality gap identified more interactions with a maximum of twenty-two on target interactions from twenty-eight different proteins when the gap is zero (Supplementary Fig. 20). All the orthogonal sets are listed in Supplementary Data 6.

Different applications require different levels of orthogonality; while gene circuits likely require extreme orthogonality, protein origami, which benefits from avidity, is not under such strict constraints. Thus, we identified the largest orthogonality gap for different numbers of on-target interactions (Fig. 4D; Supplementary Data 7). As expected, smaller sets had larger gaps, but orthogonality gaps of at least 0.5 interaction Score were identified for sets as large as seventeen on-target interactions. Finally, we compared the CCmax Library's interaction score with the iCipa predictions, which show substantial improvement over the CCNG1 Library. iCipa was able to predict interaction scores, with $R^2 = 0.43$ (Fig. 4E). We attribute the increase in iCipa's power to the use of a coiled-coil background that consists of only alanine residues at the **b**-, **c**- and **f**- positions. The improvement in predictive power appeared in other algorithms to a lesser extent, all of which maintained an $R^2 < 0.28$ (Supplementary Fig. 21).

## Discussion

We have developed and validated a system for the high-throughput identification of PPIs. We built a framework to predict orthogonal coiled-coil interactions and used it to design over 26.000 interactions, which we then assayed with the NGB2H system in a design-build-test cycle, summarized in Supplementary Data 8. Using the data collected, we improved state-of-the-art coiled-coil interaction prediction algorithms, which allowed us to design the largest set of any orthogonal proteins to date, with fifteen on-target interactions. Thus, by using iterative design, we demonstrate how high-throughput PPI characterisation can facilitate the identification of a desired protein function and improve design.

Our work builds on previous high-throughput two-hybrids to create a generalisable system for studying PPIs, which can include both soluble and membrane proteins. By uniting gene synthesis with a mapping step and a barcode readout, our system allows for the high-throughput characterisation of any binary PPI. Previous high-throughput studies used highly constrained libraries − either the ORFome[24–27] of one of a handful of reference genomes; targeted single residue mutations, which only explore a sliver of sequence space around a primary sequence[28,29], or several randomly sheared coding sequences[30]. Using the capabilities of DNA synthesis broadens the testable sequence space, which facilitates investigations of a variety of areas, such as families of protein domains, extant genetic variation, evolutionary trajectories and epistatic effects. Furthermore, for the investigator who is not interested in an all-against-all approach, synthesis allows for the explicit pairings of only certain proteins. While we benefited from the short length of our proteins of interest, recent pooled gene synthesis techniques[31,32] can be used to interrogate much larger proteins. Deconvoluting library diversity has also been a challenge for other multiplexed assays. Other multiplexed methods involved picking colonies and Sanger sequencing them[24], mapping the beginning of reading frames to reference genomes[25–27] or manually BLASTing obtained reads[30]. Our explicit mapping step allows for the

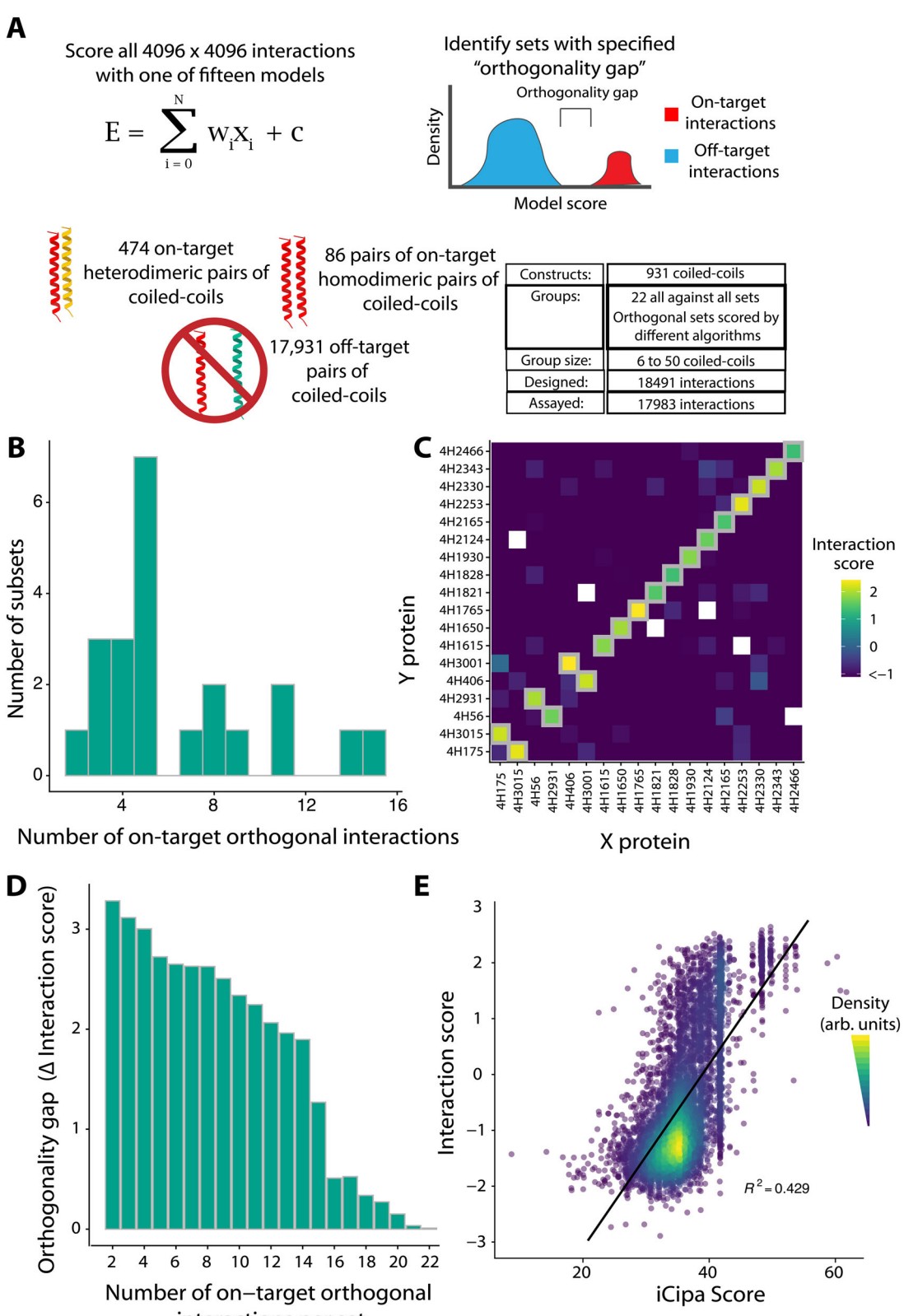

**Fig. 4 | The largest orthogonal subsets of the CCmax library. A** Design of the CCmax library. Using several iCipa variants 22 sets comprised of 18,491 interactions were designed and orthogonal sets of orthogonal interactions with a given orthogonality gap were identified (Supplementary Data 6). **B** Number of on-target orthogonal interactions per orthogonal subset at orthogonality gap of 1.0 Interaction Score. Between two and fifteen on-target, orthogonal interactions were obtained per subset. **C** The largest orthogonal subset contains fifteen on-target interactions across 318 tested pairs. Grey boxes represent designed on-target interactions. **D** The largest orthogonality gap per number on-target interactions in a set. **E** iCipa's agreement with the Interaction score ($R^2 = 0.429$). The black line is a linear model predicting interaction scores from iCipa predictions. Source data are provided as a source data file.

high-throughput creation of a library to map arbitrary proteins to DNA barcodes, and because it is a separate step, it could use long-read sequencing to overcome the length limitations of Illumina sequencing.

We do note that our system has several limitations. Notably, our system is limited to measuring dimeric interactions and unable to sense orientation and whether higher-order structures are formed. In our studies, the assemblies follow the consensus design rules and are predicted to be parallel dimeric coiled-coils[33,34], but other structures may have formed. It is also challenging to compare results between libraries. Using a next-generation sequencing readout means that each data point is relative to all other datapoints assayed at the same time, and this may change significantly between libraries that are composed of different proteins, making the comparison of interaction scores between libraries difficult. Lastly, we note that we have only tested interactions pairwise and cannot predict what might occur if more than one pair is present in solution.

Our improvements to coiled-coil design algorithms represent an important advance for de novo protein design. Though coiled-coil interactions have been modelled with diverse approaches, our iCipa algorithm shows clear advantages over existing models. In particular, heptad-shifting provides an intuitive and biologically rational addition that can be applied to any future improvements in coiled-coil design. The combination of heptad shifting with improved and novel weights for sequence features made iCipa substantially more accurate than other tested algorithms, at least for the limited set of residues tested. To increase ease of use, the iCipa scoring function is also available as a webservice at https://ajasja.github.io/icipa.

Here, we simultaneously performed a massive characterisation of PPIs within a protein family and identified the largest set of orthogonal proteins found to date. The CCmax Library characterised twice times as many total interactions as Potapov et al.[13]. From the total of 26,049 interactions we characterised, we found many orthogonal proteins — up to 15 on-target pairs at strict orthogonality gap of 1.0, which is twice the size of the largest coiled-coil set designed by Crooks et al[15]. and contains three more additional interactions than the four helix bundle orthogonal set found by Chen et al.[8]. Relaxing the orthogonality constraints produces up to 12 heterodimers or 22 heterodimers and homodimers at orthogonality gap of 0.0.

Though orthogonal coiled-coils are particularly needed as the building blocks for protein origami[4,5], they could be substituted for histidine kinases in orthogonal signalling pathways or synthetic orthogonal transcriptional logic gates[8,35,36] or for the sake of orthogonal cellular localization[37].

Thus, the ability to characterize constructs across a highly diverse sequence space and identify networked properties, such as orthogonality, highlights the NGB2H's scalability and generality. Because it can be adapted to any sequence the experimenter desires, the NGB2H facilitates the interrogation of PPIs beyond endogenous interactomes. It can be used to characterise entire protein families, empirically inform protein design, or investigate complex phenomena such as epistasis.

## Methods

### Oligonucleotide designs

Libraries were designed as shown in Supplementary Fig. 2. Though the CC0 and CC1 Libraries were assembled from two oligonucleotides and the CCNG1 and CCmax Libraries from one oligonucleotide, they followed the same overall assembly logic. In brief, each library was flanked with two orthogonal 15 bp primers[38] for amplification from the OLS pool. Interior to the flanking primers were type IIS restriction enzyme sites to facilitate scarless cloning, and the complete coiled-coil sequence. The CC0 and CC1 Libraries contain extra type IIS sites and flanking 15 bp primers to allow linking and amplification of the X and Y halves of the two-hybrid. A complete description of each design is listed in Supplementary Information Section 2 and all oligonucleotides

used are listed in Supplementary Data 1 while all proteins used are listed in Supplementary Data 2.

### Orthogonal coiled-coil interaction prediction

To predict orthogonal coiled-coils, we generated all 4,096 possible four heptad coiled-coil with asparagine or isoleucine at the **a**-position and glutamic acid or lysine at the **e**- and **g**-positions and scored 16.7 million interactions in an all-on-all design using the Potapov algorithm (CCNG1 Library) or our iCipa candidate algorithms (CCmax Library). Calculating orthogonality is a challenging problem that scales in exponential time with the number of possible binding partners. We used a maximal clique algorithm to identify sets of orthogonal coiled-coils where all on-target interactions have a higher score than all off-target interactions and it runs in under a minute on a standard laptop. Python v3.7, Pandas 0.25.1, numpy 1.15.4, jupyter lab 4.4.0 were used for the development.

### Construct and library cloning

Each library was cloned in a similar manner, with slight differences in methods to attach a random DNA barcode to the OLS pools. After the 20 bp of random DNA was attached with PCR to the 3′ end of the X and Y construct (Supplementary Fig. 3), constructs were sequenced in bulk on a MiSeq to identify it and a specific X and Y (below). After barcode mapping, the T25 and T18 + GFP halves were cloned in sequentially with type IIS restriction enzymes for scarless cloning (Supplementary Fig. 4). All enzymes and polymerases came from NEB. A complete description for how each library was cloned can be found in the Supplementary Information Section 4 and oligonucleotides used for cloning are listed in Supplementary Data 3.

### Mapping random barcodes

Once random barcodes were attached and cloned, constructs were sequenced on an Illumina MiSeq to identify the X and Y proteins which each barcode was connected to. DNA containing the X and Y proteins, and the barcode were amplified as a linear fragment, and Illumina's P5 and P7 adapters attached. Constructs were sequenced with a v3 300 cycle paired end kit (Illumina TG-142-3003), with custom primers spiked into the Illumina primers. Sequences were demultiplexed, and mapped with a BBtools pipeline and consensus building custom script. Full descriptions of how each library was mapped can be found in the Supplementary Information Section 5.

### Strains used

All NGB2H experiments were run in TK310[39] carrying pSK34. TK310 is a previously published MG1655 derivative with deletions in cpdA, lacY and cyaA, which give it a large linear response range to cAMP. pSK34 contains repressors for both the phlF and Tet promoters to maintain repression of the two-hybrid proteins. CB216 is a NEB5α derivative with pSK34 integrated genomically and only used for cloning. All plasmids used for basic cloning are listed in Supplementary Data 4 and available at Addgene. Supplementary File 2 in the Source Data contains all the plasmids used in gene bank open format.

### NGB2H assay execution

Glycerol stocks of each library were thawed, and 100uL were grown up overnight in 100 mL MOPS EZ Rich Defined Media (Teknova M2105) with kanamycin (Teknova K2125) and carbenicillin (Teknova C2130). For time course studies, a glycerol stock containing a library of constitutive GFP constructs was also thawed, and 100uL was inoculated into 10 mL of MOPS EZ Rich Defined Media with kanamycin and carbenicillin and grown overnight. The next morning 1 mL of the GFP library was added to the 100 mL of library culture. After mixing GFP and experimental libraries, 1 mL of overnight culture was added to a fresh culture of 100 mL MOPS EZ Rich Defined Media with carbenicillin and kanamycin and the inducers for two hybrid expression: 5 ng/mL anhydrotetracycline, 1.5 uM 2,4-Diacylphlorolglucinol and 100 uM

IPTG, done twice for biological replicates, except where indicated (Supplementary Information Section 6). Flasks were placed in a 37 C degree shaker for six hours. Samples were pulled after 6 h and placed on an ice slurry to quickly cool for 15 min after which cells were spun down for RNA and DNA extraction.

### RNA and DNA preparation for barcode sequencing
Samples of RNA were prepared with Qiagen RNeasy kits (Qiagen 74106, or 75144) according to manufacturer's instructions, with on-column DNase digestion (Qiagen 79254) and concentrated with RNeasy MinElute Cleanup kit (Qiagen 74204). RNA was reverse transcribed with Superscript IV (ThermoFisher 18090050) with a modified protocol such that 25 ug of input RNA was used, the extension step ran for 1 h at 55 C, and 1 uL of RNase A was added in the RNA removal step. Each sample was transcribed with a specific primer, often oSK193 or oSK194, that attached the i7 index and P7 sequencing primer. Samples of DNA were prepared with Qiagen Plasmid Plus Maxi kits (Qiagen 12963) according to the manufacturer's instructions. RNA samples were verified to contain very low levels of DNA (<1:1000) by qPCR (Kapa Biotechnology KK4601) with oSK199 and oSK200, which was repeated with a high-fidelity PCR for a low number of cycles to keep samples in the exponential amplification phase. DNA samples were similarly quantified with qPCR and amplified for low cycles to attach P5 and P7 and multiplexing indices. Amplified samples were then quantified on an Agilent Tapestation 2200 with D1000 screentape (Agilent 5067-5582), verified to be monodispersed and mixed in equimolar quantities. Complete details for RNA and DNA preparation can be found in the Supplementary Information Section 7.

### Barcode sequencing
Pooled RNA and DNA barcodes from each experiment were sequenced with various cores and startups at UCLA. The CC1 and CCNG1 Libraries were sequenced on a Hiseq 2500 while the CCmax and CC0 libraries were sequenced on a Nextseq 550. Samples were diluted and mixed with 5–20% phiX control v3 (Illumina FC-110-3001) and sequenced with oSK326 for read 1 and oSK324 for the index read.

### Barcode counting
We used a custom bash script to count DNA barcodes from barcode sequencing. After demultiplexing into reads from RNA or DNA samples, reads were truncated to the 20 bp containing the barcode and unique sequences counted. Barcode counts were then processed with Starcode (v1.3), to condense barcodes within a levenshtein distance of one to remove sequencing errors and tallied again.

### Interaction quantification
Barcode count files were imported into R where they were merged with the mapping file to provide the protein pair identified with each barcode. Barcodes corresponding to the same construct were summarized (dplyr 0.7.4) and total counts of RNA barcodes and DNA barcodes per protein pair were obtained. For our analysis we used Interactions scores calculated as $Interaction\ score = \ln\left(\mathrm{median}\left(\frac{RNA\ reads}{DNA\ reads}\right)\right)$ for barcodes that had >10 reads in all DNA samples. Interactions for all libraries are reported in Supplementary Data 5.

### Orthogonal set identification
Orthogonal sets were identified for the CCNG1 and CCmax libraries. Briefly, we wrote a script, find_orthogonal_sets_w_MIS.py that took the Interaction scores for each set and built a graph with interactions forming the edges between proteins. Finding the maximum independent set of the line graph of this graph gave us the largest orthogonal set of interactions, listed in Supplementary Data 6. The largest sets for different numbers of on-target interactions are listed in Supplementary Data 7. Supplementary File 1 in the Source Data contains excel files for all designed sets, the paring of peptides and a heat plot map.

### Statistics and reproducibility
We have verified that technical repeats are very well correlated (Fig. 1F) and so sequencing experiments were only done once, unless explicitly stated otherwise in the main text. No data was excluded. No statistical method was used to predetermine sample size. Biological replicates were performed for CC0 library and all the CC0 library was included in all sequencing experiments as an internal control. The experiments were not randomized. The developed scoring function was tested on novel experimental data. The Investigators were not blinded to allocation during experiments and outcome assessment.

### Reporting summary
Further information on research design is available in the Nature Portfolio Reporting Summary linked to this article.

## Data availability
The raw sequencing data generated in this study have been deposited in the Sequence Read Archive under accession code PRJNA737455. The processed data (computational scores, set predictions and processed interactions scores) are available at https://github.com/ajasja/NGB2H and on Zenodo with the https://doi.org/10.5281/zenodo.7774717. Plasmids pSK33, 34, 59, 168 and 179 are available in the Addgene repository (https://www.addgene.org/) under accession codes 193731, 193732, 193736, 193737, 193738 and 196340. Source data are provided with this paper.

## Code availability
Code needed to design orthogonal sets, score CC pairs, process data and reproduce figures in the main deposited text is available at https://github.com/ajasja/NGB2H. The exact version of the code used is also available on Zenodo with the https://doi.org/10.5281/zenodo.7774717. The iCipa scoring function is also available as a webservice at https://ajasja.github.io/icipa.

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

## Acknowledgements

We thank the members of the Kosuri and Plesa labs for their feedback on the manuscript and figures. We thank Suhua Feng of the UCLA Broad Stem Cell Research Center and the team of the Technology Center for Genomics and Bioinformatics for performing the next-generation sequencing. We thank Octant Inc., the Kruglyak Lab at UCLA and the Black Lab at UCLA for the use of their next-generation sequencers. We thank Mathew Graf and Will Silkworth for their assistance at the UCLA-DOE Biochemistry Shared Instrumentation Facility. We thank Thomas Kuhlman for kindly providing strain TK310. We thank Amy Keating for sharing the Potapov CC scoring scripts. Finally, we thank Chris Voigt for sharing repressor/promoter sequences with us. This work was supported by the following funding sources: The National Institutes of Health (DP2GM114829) to S.K., Searle Scholars Program to S.K., ERA-SynBio (1445112) to S.K. and R.J. European Union's Horizon 2020: CC-LEGO 792305 to A.L., ERC project MaCChines (787115) to R.J. and FET Open project Virofight (899619) to R.J. and Slovenian Research Agency projects: CC-TRIGGER J1-4406 to A.L., P4-0176 to R.J. and J1-9173 to R.J.

## Author contributions

H.K., W.C.B., N.L. and S.K. designed the NGB2H system. A.L., D.S. and R.J. designed the large sets of coiled-coils. H.K., N.L. and W.C.B. designed the oligonucleotide libraries. H.K. W.C.B. and J.L. performed the experiments. A.L. designed the improved interaction algorithms. W.C.B. and N.L. performed the computational analysis. W.C.B, A.L., R.J. and S.K. analysed the results and iteratively planned the next steps. W.C.B created the figures. S.K., W.C.B. and A.L. wrote the manuscript, with input from all authors.

## Competing interests

S.K. is cofounder and CEO and holds equity, N.L. is an employee and holds equity and J.L. was an employee and holds equity in Octant Inc. All other authors declare no competing interests.
