## [Peer Review File · Nature Communications]

Reviewers' Comments:

Reviewer #1:

Remarks to the Author:

The paper seeks to explore dimeric coiled coil interactomes coupled with a bacterial Two-hybrid screen to provide quantitative data on the the interactions observed within the dataset. The B2H system allows the potential PPIs to be screened within the cell, with the assay readout simplified by coupling to an elegant barcode readout.

Next the authors assay potential interactors from the dataset in order to build and train a coiled coil based scoring system, arriving at a set of 12 orthogonal heterodimeric interactions. This is approximately four more than has previously been reported for a heterodimeric set.

The science presented here is very nice, detailed, and the analysis is carefully considered. The paper would benefit from a section/table consolidating a clear description outlining the various libraries, their differences (CC0, CC1, ccMax, CCNG1), the various algorithms used, and discussing key details and differences on how they function and their levels of accuracy. This would really help to highlight the modest improvements that are achieved by the resulting icipa algorithm.

Reviewer #2:

Remarks to the Author:

This work presents two interesting things, each with the potential to be important contributions: a quantitative high-throughput screen for protein interactions encoded by (short) synthetic genes, and several large sets of coiled coils that form various patterns of interactions, identified using the screen. The screen is impressive, validated by extensive controls, and has some very nice technical aspects. Notably, it gives readouts in good agreement with previously measured thermal stabilities for a set of ~25 coiled coils. Sets of orthogonal coiled coils are in demand for a variety of protein engineering and synthetic biology applications, so this result is also an important contribution, though there are unresolved questions about how orthogonal the presented interactions are, as detailed below.

In its current form, the paper is unpolished and accessible only to readers who already know quite a bit about coiled coils, high-throughput screening, and this specific design problem. The paper does not provide enough context for most readers. Also, key terms and concepts need to be defined, and more background on past work, particularly the older models that are used in the paper, is needed. The paper is hard to read and difficult to understand in some of its specifics, due to sentences with unclear/imprecise meanings.

Major points:

- The minimum difference in affinities between on- vs. off-target interactions is presumably what the authors are labeling the "orthogonality gap" (though this should be clearly defined). What are the experimental orthogonality gaps for the sets in Figure 2B, C and the large set of 22 orthogonal pair in Figure 4C? These do not appear to be large. If the gaps are not larger than the estimated experimental noise, then I don't see how these sets of interactions can be labeled as "orthogonal."
- Related to this, Table S7 is cited as the source of data in Figure 4D, but the figure and the data file don't agree. Table S7 lists negative orthogonality gaps for 10 of 22 sets. If the orthogonality gap is negative, why is it shown as positive in Figure 4D and why should these sets be called "orthogonal?" Any set referred to as orthogonal should be associated with a positive gap that is greater than the estimated experimental noise.
- Figures 1E and H and Figure S8 D indicate fairly large RMS errors (many points with errors ≥ 1 unit) arising from which fusions are used. This provides one measure of error/uncertainty. This magnitude of error suggests that sets with orthogonality gaps of less than ~ 1 unit aren't confidently orthogonal (and based on Figure 4D and Table S7 this is the large majority of the presented sets). Particularly in the absence of a validating assay, an estimate of the assay error and how it corresponds to the orthogonality gap is needed so that users can assess the potential utility of these proteins for their applications.
- Validation using another assay would make these sets of proteins much more valuable to labs

that might want to use these new tools.

- The authors don't discuss the possibility that some - perhaps many - of their pairs form structures other than parallel homo/heterodimers. Higher-order oligomers or antiparallel structures could be formed. Does the assay return a positive signal for antiparallel control peptides? This would be easy to test using sequences from the Jerala lab (or other known antiparallel coiled coils) and will be very important to know for most applications that might use these components, including protein origami.

Other points:

- Are correlation coefficients (e.g. in Figure 1) computed using all data points, including the frameshifted/indel non-interactions? If so, the points that don't encode full-length proteins should be omitted.
- Figure 2E is not convincing. How well is the shape of the spline determined by the data?
- In the CCmax part of the paper it is unclear whether the sets that are referred to are the sets that were designed, or are instead sets that were extracted after the fact by mining the experimental interactions. (Wording: "We identified orthogonal sets...")
- Related, it is unclear to what extent the design process helped identify orthogonal sets, vs. just screening a lot of pairs and then identifying orthogonal sets from them. I didn't understand Figure S18 or the corresponding main text (what is the subsampling analysis for?).
- Previously published scoring models aren't described sufficiently for non-experts to understand why they are being used in this work, and they are not presented in the appropriate context in terms of what they should be able to do. These models take two sequences - and a user-specified alignment - and return a score. They can't predict an interaction score without a user-provided alignment. These methods can and should be used to score shifted alignments, to find the best score, as is done here with "heptad shifting." The observation that heptad shifting improves the correlation with data in this study is a nice result. But scoring different alignments isn't a "new method." The impact of shifting on predictions using the existing models should also be tested.
- If (all) models are showing poor performance because some structures aren't parallel dimers, then re-training using data from this screen won't help. I suspect improvements in R2 on CC0 may come from over-training on closely related sequences, which is hard to avoid in this very limited space. This could perhaps be addressed by partitioning the testing and training sets by some measure of similarity (though I'm not sure it's a worthwhile analysis, since none of the methods do very well).
- Why does the scale of the interaction scores vary so much between libraries? Can scores be compared between experiments/libraries?
- There should be an easy-to-read supplementary figure or table that shows all of the interacting sequences in some of the orthogonal sets, esp. the highlighted sets, in aligned pairs as predicted by the authors' model. Ideally, these would be color-coded, so that readers can see the features present in interacting and non-interacting pairs.
- Given that the mapping is so key to the screen, I recommend incorporating an overview of Figure S4 into the main text Figure 1.
- The very significant work of Zibo Chen (ref 8), who did the key orthogonality test of mixing all peptides in a single pot, for 16 designed pairs, is cited but not discussed. This is more "state of the art" than the frequently mentioned Crooks data.
- Related, the authors are advised to caution readers that they do not know what will happen when they mix >2 of their peptides together, since all interactions are tested pairwise.

Minor points:

- In J. Mol. Bio. 1998 Vinson reported favorable coupling energies between Lys residues at e and g, relative to Ala-Ala.
- Referring to possible applications of orthogonal dimers in histidine kinase signaling is not related to reference 30, which is cited.
- CCNG1 = CC1?
- There are lots of sentences where I'm not sure exactly what is meant, or what is being referred to (one example: "20 of our 51 sets" on p. 7 is puzzling - what 51 sets are being referred to? I'm guessing the analysis described in the previous paragraph led to 51 orthogonal sets? The reader has to fill in a lot of gaps here and in other places; writing needs to be more precise).
- The Supplementary Tables need to be index/labeled. The excel files have cryptic names; it was hard to find information.

- Coiled coil notation traditionally uses lowercase letters to refer to positions a...g; notation in this paper is not consistent and sometimes uses uppercase
- There are lots of typos

Comment: it could be that coiled coils made with the non-Ala b,c,f backgrounds (which in my opinion would be better terminology than "backbones" as used in the text) give higher interaction signal than the mostly-Ala background because the Ala patches promote non-specific interactions in the cellular environment that compete with enzyme complementation. I.e. absent validation in a purified system, it's not clear that this is a feature of the interactions themselves.

We thank both reviewers for the kind and thorough comments on our manuscript. We have attempted to address the concerns raised here and improved the clarity of the manuscript as suggested. We also apologize for the length of time in returning this manuscript back to reviewers; both lead authors moved on as well as the last author whose academic laboratory shut down and everything happened in the middle of a global pandemic. We do believe the manuscript provides a valuable reference in the field, by describing a new experimental and computational technique and as a large resource for the synthetic biology community for more sets of orthogonal peptide interactors. Please find our point-by-point responses below, with the original comments in gray and our responses in bold. Text in quotation marks (“”) refers to text taken verbatim from the improved manuscript.

We have also set up a webserver at <https://ajasja.github.io/icipa>, that enables any researcher to try out the iCipa scoring function. The code needed to design orthogonal sets, score CC pairs, process data and reproduce figures in the main text is available at <https://github.com/ajasja/NGB2H>. We have also added visualizations for each design set to the repository as well as a supplementary file S1.

Point by point reply to reviewer comment

Reviewer #1 (Remarks to the Author):

The paper seeks to explore dimeric coiled coil interactomes coupled with a bacterial Two-hybrid screen to provide quantitative data on the the interactions observed within the dataset. The B2H system allows the potential PPIs to be screened within the cell, with the assay readout simplified by coupling to an elegant barcode readout.

Next the authors assay potential interactors from the dataset in order to build and train a coiled coil based scoring system, arriving at a set of 12 orthogonal heterodimeric interactions. This is approximately four more than has previously been reported for a heterodimeric set.

The science presented here is very nice, detailed, and the analysis is carefully considered.

We would like to thank the reviewer for the encouraging comments!

The paper would benefit from a section/table consolidating a clear description outlining the various libraries, their differences (CC0, CC1, ccMax, CCNG1), the various algorithms used, and discussing key details and differences on how they function and their levels of accuracy. This would really help to highlight the modest improvements that are achieved by the resulting icipa algorithm.

We have added a table (Table S8) outlining the various libraries that were designed and tested and what the improvement in the size of the orthogonal sets were. The accuracy of the improvements of iCipa are demonstrated in Figure 3 but can also be seen from the increasing size of the orthogonal sets (column “Largest experimental orthogonal set” in Table S8). The improvement of iCipa is also shown in supplementary figure S22, showing that iCipa has the best correlation with the CCmax library experimental data of all the tested models.

Library	Purpose of library	Algorithm used in design	Size of library	Largest experimental orthogonal set found (gap>0)	Largest experimental orthogonal set found (gap>0.5)	Largest experimental orthogonal set	Alias/working name
CC0	To validate and test the NGB2H system	Using bCipa as the scoring function, pairs, quadruples and octuples were found as described in Crooks et al, 2017	256 interaction pairs in one 16x16 set	7 pairs	N/A	N/A	mason
CC1	Control library intended to explore the effects of different backgrounds (sets of amino acids at the b,c,f positions)	The peptides were designed by hand in the Jerala lab (https://doi.org/10.1002/psc.1331) and the effects of substitutions at the background positions have been extensively characterized in (https://pubs.acs.org/doi/10.1021/jacs.7b01690) and used in vivo in (https://www.nature.com/articles/s41589-019-0443-y)	400 interaction pairs in a 20x20 set.	N/A	N/A	N/A	R20
CCNG1	First library to test the orthogonal design pipeline. Included several controls and several different residues at the b, c, f positions (referred as backgrounds in the text).	Using the Potapov scoring function all possible combinations of I/N at a position an EK at g and e positions were scored and orthogonal sets identified as a variant of the maximum independent set problem.	8169 interactions pairs in 56 sets containing between 64 to 961 interactions.	12 pairs	8 pairs	6 pairs	R8000
CCmax	First library design using the new iCipa scoring function. To reduce the number of pairs, only a single background (set of b, c, f) positions was used.	The iCipa scoring function was developed based on the data from the CCNG1 library. iCipa was used to score all possible combinations of I/N at a position an EK at g and e positions were scored and orthogonal sets identified as a variant of the maximum independent set problem.	18491 interaction pairs in 15 orthogonal sets and seven control sets.	22 pairs	16 pairs	15 pairs	R18000

We have also added the following text:

We have developed and validated a novel system for high-throughput identification of PPIs. We built a framework to predict orthogonal coiled-coil interactions and used it to design over 26,000 interactions which we then assayed with the NGB2H system in a design-build-test cycle, summarized in Table S8.

Reviewer #2 (Remarks to the Author):

This work presents two interesting things, each with the potential to be important contributions: a quantitative high-throughput screen for protein interactions encoded by (short) synthetic genes, and several large sets of coiled coils that form various patterns of interactions, identified using the screen. The screen is impressive, validated by extensive controls, and has some very nice technical aspects. Notably, it gives readouts in good agreement with previously measured thermal stabilities for a set of ~25 coiled coils. Sets of orthogonal coiled coils are in demand for a variety of protein engineering and synthetic biology applications, so this result is also an important contribution, though there are unresolved questions about how orthogonal the presented interactions are, as detailed below.

In its current form, the paper is unpolished and accessible only to readers who already know quite a bit about coiled coils, high-throughput screening, and this specific design problem. The paper does not provide enough context for most readers. Also, key terms and concepts need to be defined, and more background on past work, particularly the older models that are used in the paper, is needed. The paper is hard to read and difficult to understand in some of its specifics, due to sentences with unclear/imprecise meanings.

We would like to thank the reviewer for seeing the promise of the work, for taking the time to help us improve the presentation and accessibility. We feel the resulting manuscript is much improved as a result.

Major points:

- The minimum difference in affinities between on- vs. off-target interactions is presumably what the authors are labeling the “orthogonality gap” (though this should be clearly defined). What are the experimental orthogonality gaps for the sets in Figure 2B, C and the large set of 22 orthogonal pair in Figure 4C? These do not appear to be large. If the gaps are not larger than the estimated experimental noise, then I don’t see how these sets of interactions can be labeled as “orthogonal.”

The inferred definition of the orthogonality gap is indeed correct, we have more explicitly defined it in the text (“orthogonality gap is calculated as the weakest on-target interaction minus the strongest off-target interaction score”).

In Fig 2B (4h-1or2N-not first only B07 bc-8.05 nc-7.05-all.00) the experimental orthogonality gap is 0.14 in Fig 2C (4h-1or2N-not first only B07 bc-8.05 nc-7.05-hetero-ex.01.00) the gap is 0.11 and in Fig 4C (4H 1or2N 5 DNA-ALL-basicL-rep-core vertical-Ridge-WbnRD10 P07 bc-48.00 nc-42.00-hetero-ex.00.00.aA) the orthogonality gap is 0.01.

Depending on the specific application a low orthogonality gap might be acceptable (for example in building coiled coil protein origami). In Table S6 we have prepared sets with larger orthogonality gaps (0.5 and 1.0) for more stringent applications.

- Related to this, Table S7 is cited as the source of data in Figure 4D, but the figure and the data file don’t agree. Table S7 lists `_negative_` orthogonality gaps for 10 of 22 sets. If the orthogonality gap is negative, why is it shown as positive in Figure 4D and why should these sets be called “orthogonal?” Any set referred to as orthogonal should be associated with a positive gap that is greater than the estimated experimental noise

The Table S7 included with the manuscript was inadvertently included from a previous analysis. We apologize for the oversight and have corrected it, while incorporating improvements in our analysis regarding Figure 4D.

- Figures 1E and H and Figure S8 D indicate fairly large RMS errors (many points with errors ≥ 1 unit) arising from which fusions are used. This provides one measure of

error/uncertainty. This magnitude of error suggests that sets with orthogonality gaps of less than ~1 unit aren't confidently orthogonal (and based on Figure 4D and Table S7 this is the large majority of the presented sets). Particularly in the absence of a validating assay, an estimate of the assay error and how it corresponds to the orthogonality gap is needed so that users can assess the potential utility of these proteins for their applications.

A measure of uncertainty can be gained from looking at the standard deviation (RMSD) of the difference between the two biological replicates in Fig 1C. The RMSD value is 0.37 interaction score units.

We have analyzed several different orthogonality gaps, depending on the requirements of the application. For users looking for a more conservative approach we have updated our analysis to identify sets with orthogonality gaps of Interaction Score 0.0, 0.5 and 1.0. These sets are also reported in Table S6. Orthogonal sets with different orthogonality gaps are also shown in Figures S12 and S20. We have added the following text:

A measure for the assay uncertainty can be estimated from the standard deviation of the difference between the two biological replicates (RMSD) in Fig 1C. The value of the RMSD is 0.37, so for applications that require very high levels of orthogonality an orthogonality gap of 0.5 is recommended. For constructing protein origami assemblies an orthogonality gap of 0.0 is sufficient as demonstrated by Aupič et al.²¹, where two identical pairs were used in the same structure. Pairwise orthogonality is the most stringent criterion. In a single pot experiment, where all pairs would be present, we speculate that orthogonality would only improve, since the off-target states would be competing with the on-target states.

We have also investigated a more stringent measure of uncertainty. Each of the tested libraries contained the CC0 set as an internal control. We have mapped the experimentally measured CC0 Tm temperatures to interaction score units using a linear fit. We have calculated an RMSD between the interactions scores from Tm and the directly measured interactions scores and used this RMSD as the orthogonality gap. We have added the following text.

As the orthogonality gap of these interactions is not large, we calculated similar results using orthogonality gaps of 0.5 Interaction score and 1.0 Interaction score. We have also mapped back the previously measured melting points of CC0 to interaction scores using a linear fit (Figure S10C) and calculated the RMSD between melting point interaction scores and the experimental CC0 interactions scores measured as part of CCNG1. The RMSD was approximately 0.7 interaction score units. All sets are presented in Table S6.

- Validation using another assay would make these sets of proteins much more valuable to labs that might want to use these new tools.

We agree that validating orthogonal sets with additional assays would provide additional confirmation (as for any study). Unfortunately, the Kosuri Lab is now shut down, and the lead authors have moved onto other pursuits. We feel that the paper already contains a substantial amount of results, and novel methods that are valuable to the scientific community and should be reported. We have

received communications from other researchers who have already started to use the sequences of new pairs provided in the bioRxiv version.

- The authors don't discuss the possibility that some - perhaps many - of their pairs form structures other than parallel homo/heterodimers. Higher-order oligomers or antiparallel structures could be formed. Does the assay return a positive signal for antiparallel control peptides? This would be easy to test using sequences from the Jerala lab (or other known antiparallel coiled coils) and will be very important to know for most applications that might use these components, including protein origami.

We added a section to our discussion where we address the possibility of higher order oligomers, (“Notably our system is limited to measuring dimeric interactions and is unable to sense whether higher order structures are formed. In our studies the assemblies follow the consensus design rules and are predicted to be parallel dimeric coiled-coils^{31,32”}).

We note that according to the consensus on the rules guiding coiled-coil association our designs should be primarily dimeric as they invariantly have Leu at the d-position and generally containing some Ile at the a-position. With respect to CC orientation, we have in fact tested two antiparallel sets in the CCmax library: a control containing the AP1-AP12 which were used in Fink et al.

(<https://www.nature.com/articles/s41589-018-0181-6>) and a newly designed set.

We found they most of them did not give appreciable signal (with the exception of the AP5-AP6 interaction). We postulate that in order to detect antiparallel binding in NGHB2 assay, the linkers to the split cAMP enzyme might have to be further optimized.

Other points:

- Are correlation coefficients (e.g. in Figure 1) computed using all data points, including the frameshifted/indel non-interactions? If so, the points that don't encode full-length proteins should be omitted.

All correlations and other analyses are computed only using constructs where both proteins have sequences that perfectly match the intended designed sequences (thus synthetic errors are not analyzed in this work). We have amended the text to reflect this. (“Only barcodes for which ten or more reads were obtained in every DNA replicate and that perfectly mapped to designed protein pairs were used in further analysis.”)

- Figure 2E is not convincing. How well is the shape of the spline determined by the data?

We have changed this figure to use a polynomial of degree two to facilitate interpretation and have included a statistical analysis to show the appropriateness of using polynomial regression.

- In the CCmax part of the paper it is unclear whether the sets that are referred to are the sets that were designed, or are instead sets that were extracted after the fact by mining the experimental interactions. (Wording: “We identified orthogonal sets...”)

We have changed the text (“Similarly to the CCNG1 library, we identified the largest orthogonal subsets of each designed set”) to make it clearer that this identification was done in the same manner as the CCNG1 set by identifying smaller orthogonal sets from the designed orthogonal sets.

- Related, it is unclear to what extent the design process helped identify orthogonal sets, vs. just screening a lot of pairs and then identifying orthogonal sets from them. I didn’t understand Figure S18 or the corresponding main text (what is the subsampling analysis for?).

We thank the reviewer for pointing unclear sections of the text. We have added this paragraph at the end of section “Computational design of large sets of orthogonal coiled-coils”:

The space of all possible pairs, assuming only our limited set of amino acid residues (~16 M) is several orders of magnitude larger than what could be screened experimentally (~20 k), so the design process is crucial for identifying feasible orthogonal sets that can be experimentally tested.

The subsampling analysis presented in Figure S18 was meant to illustrate the difficulties in designing orthogonal sets due to score function differences. The same orthogonal set scored with a different scoring function shows varying levels of orthogonality. But since it is not essential and might cause confusion we have decided to remove it and the corresponding text.

- Previously published scoring models aren’t described sufficiently for non-experts to understand why they are being used in this work, and they are not presented in the appropriate context in terms of what they should be able to do. These models take two sequences – and a user-specified alignment – and return a score. They can’t predict an interaction score without a user-provided alignment. These methods can and should be used to score shifted alignments, to find the best score, as is done here with “heptad shifting.” The observation that heptad shifting improves the correlation with data in this study is a nice result. But scoring different alignments isn’t a “new method.” The impact of shifting on predictions using the existing models should also be tested.

We have added a paragraph describing previously published scoring methods at the beginning of the section “Computational design of large sets of orthogonal coiled-coils”:

All dimeric coiled-coils have a similar structure, which is why sequence based scoring functions can fruitfully predict melting temperatures or binding affinities. The scoring functions accept as input two sequences, usually starting with a specific register and return a score. One of the widely used algorithms is bCipa¹⁴ which is based on summing weights for residue-residue

interaction pairs as well as electrostatic interactions and helical propensity and predicts melting temperatures. The state-of-the-art scoring function was developed by Potapov *et al.*¹³, which uses triplet weights in addition to the pair weights and a much larger training set to predict the free energy of binding. The paper also benchmarks the most common CC scoring functions such as Fong/SVM²⁰ and Vinson/CE¹².

While previous algorithms can be viewed as separate from the alignment, every paper we have seen which uses these algorithms (for example in Potapov *et al.*, PLOS, 2015 or in Crooks *et al.*, Biochemistry, 2017), presumes the full alignment of two coiled-coils and does not consider shifting.

We have tested the impact of the shifting using bCipa and the Potapov scoring function and added the results to Figure S16. (“The effect of heptad shifting on iCipa as well as bCipa and the Potapov scoring function is shown in Figure S16.”) Both methods show a slightly better correlation when heptad shifting is performed.

- If (all) models are showing poor performance because some structures aren't parallel dimers, then re-training using data from this screen won't help. I suspect improvements in R2 on CC0 may come from over-training on closely related sequences, which is hard to avoid in this very limited space. This could perhaps be addressed by partitioning the testing and training sets by some measure of similarity (though I'm not sure it's a worthwhile analysis, since none of the methods do very well).

It is indeed possible that iCipa has been over-trained on sequences closely related to the CC0 library. However, all ~2000 bA background interactions were used to train iCipa. As our library is not composed to be near in sequence space to the CC0 library, it is not expected that a model drawn from our data should have an improved predictive ability of the CC0 library. However, since we are using a limited sequence space, some sequence similarity is unavoidable. The iCipa scoring function also performs very well on the CCmax library on which it has not been trained (Figure S22)

- Why does the scale of the interaction scores vary so much between libraries? Can scores be compared between experiments/libraries?

Although it is potentially possible to compare scores between different libraries this a relatively complex analysis and should be treated with caution. Each score can only be considered in the context of the other scores it was measured with. Theoretically a function could be fit between two subsets that are shared between libraries but is not recommended. While the relationship is presumably monotonic it may not be linear. We have added a section to our discussion on this issue (Using a next-generation sequencing readout means each data point is relative to all other datapoints assayed at the same time and this may change significantly between libraries that are composed of different proteins, making comparison of interaction scores between libraries difficult”).

- There should be an easy-to-read supplementary figure or table that shows all of the interacting sequences in some of the orthogonal sets, esp. the highlighted sets, in aligned pairs as predicted by the authors' model. Ideally, these would be color-coded, so that readers can see the features present in interacting and non-interacting pairs.

We thank the reviewer for the suggestion. We have added Excel and interactive HTML files to the supplementary material, which provides interactive descriptions of the on-target interactions and other metrics of the set. These are also available at <https://github.com/ajasja/NGB2H> in the folder 06_set_visualizations.

Here are some examples with clickable links: sets in Fig2B, Fig2C and Fig4C. Images are shown below:

An interactive plot of the CCNG1 library score vs iCipa prediction without and with shifting have also been made (https://github.com/ajasja/NGB2H/tree/main/06_set_visualizations).

- Given that the mapping is so key to the screen, I recommend incorporating an overview of Figure S4 into the main text Figure 1.

Thank you for the suggestion. A trimmed down version of S4 is now included in Figure 1A.

- The very significant work of Zibo Chen (ref 8), who did the key orthogonality test of mixing all peptides in a single pot, for 16 designed pairs, is cited but not discussed. This is more “state of the art” than the frequently mentioned Crooks data.

We agree that the work of Chen et al. should be given more attention and have added a short discussion into the introduction.

In a state-of-the-art example Chen *et al.*⁸ performed a single-pot experiment mixing fifteen designed heterodimer pairs, which resulted in a set of 12 orthogonal heterodimers. However, the Rosetta energy function did not successfully predict orthogonality and predicting orthogonal binding and designing large orthogonal sets remains beyond current *de novo* design methods³.

The proteins tested in Chen et al. were not designed to be orthogonal (unless one counts the implicit negative design conferred by buried HB-nets). In fact, the Rosetta scoring function is not capable of predicting the off-target interactions for this type of protein, although this is partially also a sampling problem, since the structure of the off-target interactions are not known.

The peptides tested by native mass spectroscopy in the single pot experiment were chosen based on the variability of the buried HB-nets. As a detail, in the cited report there were 15 different pairs tested (37_ABXB¹⁵N_ is the same proteins as 37_ABXB, just labeled with heavier nitrogen atoms) and the orthogonal subset obtained from the experiment contains about 11-13 pairs (depending on the orthogonality gap one chooses to use). Additionally, Chen et al mention that noncognate trimers and higher oligomers were observed but only dimers were filtered by MS and selected for presentation.

- Related, the authors are advised to caution readers that they do not know what will happen when they mix >2 of their peptides together, since all interactions are tested pairwise.

Thank you and we have updated the discussion to reflect this caveat (“Lastly, we note that we have only tested interactions pairwise and cannot predict what assemblies may occur if more than one pair is present in solution.”).

Minor points:

- In J. Mol. Bio. 1998 Vinson reported favorable coupling energies between Lys residues at e and g, relative to Ala-Ala.

We thank the reviewer for bringing this to our attention and have incorporated it into our text (“Perhaps counterintuitively, Lys/Lys pairings are acceptable, and previous biochemical work has identified mildly favorable binding contributions for Lys/Lys pairings²¹”)! This is an additional validation of iCipa model, since positive Lys-Lys weights have been observed in all iCipa models we have trained.

- Referring to possible applications of orthogonal dimers in histidine kinase signaling is not related to reference 30, which is cited.

We had originally referred to this having in mind orthogonality in signaling, but you are correct and we are removing the reference.

- CCNG1 = CC1?

CC1 is another control library, containing peptides P1-P12 from the Jerala lab, that was used to test the effect of different backgrounds on the interaction score and is presented in Fig S8. We have clarified all the libraries in Table S8. We have made the reference to CC1 clearer in the text (“After testing a subset of the CCNG1 Library to validate our in-house designs, which we call the CC1 Library”).

- There are lots of sentences where I’m not sure exactly what is meant, or what is being referred to (one example: “20 of our 51 sets” on p. 7 is puzzling – what 51 sets are being referred to? I’m guessing the analysis described in the previous paragraph led to 51 orthogonal sets? The reader has to fill in a lot of gaps here and in other places; writing needs to be more precise).

We thank the reviewer for improving the manuscript. We have clarified our terms by explicitly defining them in the text, notably:

- **orthogonality gap** ("orthogonality gap is calculated as the weakest on-target interaction minus the strongest off-target interaction."),
- **designed orthogonal set, on-target and off-target** (“For example, designed orthogonal sets of interacting proteins, where each pair of proteins interacts only with its intended on-target pair and none of the other off-target proteins present in the set...”)
- **orthogonal subset** ("Thus, we refer to an *orthogonal subset* as the largest experimentally characterized group of orthogonal interactions from what was computationally predicted to be an orthogonal set.”)

- The Supplementary Tables need to be index/labeled. The excel files have cryptic names; it was hard to find information.

Thank you for the comment. We have updated the supplementary tables to have clearer names and added the description of the tables to the supplementary materials. The tables are now named:

Table S1 OLS oligonucleotides for gene synthesis of libraries
Table S2 Sequences of proteins used in libraries
Table S3 Oligonucleotides for PCR
Table S4 Plasmids referred to by name
Table S5 Interaction Scores from all libraries
Table S6 Largest orthogonal subset for each set in the CCMAX library
Table S7 Largest orthogonality gaps by number of on-target interactions
Table S8 Comparisons between libraries

We have also added two supplementary files:
Supplementary File S1 Visualization of designed set
Supplementary File S2 plasmid files in GB format

- Coiled coil notation traditionally uses lowercase letters to refer to positions a...g; notation in this paper is not consistent and sometimes uses uppercase

We thank the reviewer for pointing this out and have updated the figures and supplement where the heptad positions were capitalized.

- There are lots of typos

We have attempted to correct these wherever found and have also had the main article text proofread by a professional academic proofreading company.

Comment: it could be that coiled coils made with the non-Ala b,c,f backgrounds (which in my opinion would be better terminology than “backbones” as used in the text) give higher interaction signal than the mostly-Ala background because the Ala patches promote non-specific interactions in the cellular environment that compete with enzyme complementation. I.e. absent validation in a purified system, it’s not clear that this is a feature of the interactions themselves.

We would like to thank the reviewer for suggesting the term backgrounds! We have adopted the term throughout the paper.

Based on the results of Drobnak et al (<https://pubs.acs.org/doi/abs/10.1021/jacs.7b01690>), where a model relating the strength of intramolecular charge interactions and helical propensity was developed and on the results of Lebar et al (<https://www.nature.com/articles/s41589-019-0443-y>) which used mostly-Ala backgrounds but still obtained high orthogonality *in vivo* we feel confident that that the backgrounds are not likely to cause aggregation. If the mostly alanine-backgrounds were agglomerating in the cellular environment and being sequestered there, we would anticipate that the alanine-backgrounds would have lower on-target interaction scores as well. We would like to point out that alanine heavy backgrounds have among the highest on-target interaction scores we

measured, which fits with the model of Drobnak et al, since the Ala residues strongly increase the helical propensity of the peptides.

Reviewers' Comments:

Reviewer #2:

Remarks to the Author:

The manuscript by Boldridge is easier to read and understand and some of my comments were addressed. However, the article continues to refer to sets of protein interactions as "orthogonal" if the strongest off-target interaction has a lower score than the weakest on-target, even if this difference is tiny and experimentally insignificant. The set of proteins that is the highlight of the paper, shown in Figure 4C and referred to in the last sentence of the abstract (... the largest set of orthogonal PPIs to date...), has an orthogonality gap of just 0.01. I asked about this value in my review, and the authors provided it in their rebuttal. It is also available in the manuscript, although it is not emphasized or easy to find. The authors state the estimated error of the assay as ~ 0.37 interaction score units, which I think is overly optimistic (see below). But even this estimate makes a difference of 0.01 negligible. The authors have a considerable amount of other data that point to larger uncertainty, particularly the error that they get when reversing the fusion pairs (swapping proteins "1" and "2" – in Figure 1E - or "X" and "Y" – Figure 1H and S17, where no RMS error is reported, only a correlation coefficient) and the error when they re-run the CC0 library repeatedly (Figure S18; other analyses could also be done using the CC0 replicates to add rigor). There are many tests that the authors could do, using the data they already have, to estimate the error more appropriately and then apply this as a reasonable cutoff for an orthogonality gap. Even in the absence of further experimental validation, this could provide some reasonable support for new orthogonal sets. They don't do this, however, and as a result, the labeling of many of the sets presented here as "orthogonal" is not justified. Although the processed data are available as part of the paper, and researchers using these data can use whatever orthogonality cutoff they prefer, the use of unjustified cutoffs to define orthogonal sets that are a major claim of the paper is not appropriate.

When looking into the reproducibility, I noticed that sometimes the interaction matrices are reported as full matrices (Figures 1H, 4C, S20) and sometimes as half matrices (Figures 2B and C). In Figure 1H, there is noise when comparing the reciprocal interactions, as expected. Figure S17E suggests there should be quite a lot of noise. But to the eye, Figure 4C appears perfectly symmetrical. This is highly implausible. If the matrix was symmetrized, then it is very misleading to show the whole thing. The experimental asymmetry provides a highly relevant measure of expected uncertainty that would be appropriate to use in the analysis, but it seems not to be included in Figure 4C. It looks like the matrices in Figure S20 have also been symmetrized.

I agree with the authors that providing these data and models to the community will make a meaningful contribution. There are many interesting methods and results in this work. I also recognize that in their current circumstances, they can't do more experiments. However, in its current form, I think the manuscript is misleading and makes claims about orthogonality that are not supported by the data. Also, I will again raise Figure 2E, which I continue to find unconvincing despite the authors' rebuttal. A fit to a polynomial that is statistically better than a fit to a line, when both fits are extremely poor and the trend is not well-defined by available data, does not provide good support for their claim. Finally, although not a criterion for acceptance (since the sequences are all given) it seems that the authors are not making the clones available (e.g., in AddGene), which is regrettable.

We thank the reviewer for the additional comments and for taking the time to help us improve our manuscript.

Please find our point-by-point responses below, with the original comments in gray and our responses in **bold**. Text in quotation marks (“”) refers to text taken verbatim from the improved manuscript.

Point by point reply to reviewer comments

Reviewer #2 (Remarks to the Author):

The manuscript by Boldridge is easier to read and understand and some of my comments were addressed.

We would like to thank the reviewer for the encouraging comments!

However, the article continues to refer to sets of protein interactions as “orthogonal” if the strongest off-target interaction has a lower score than the weakest on-target, even if this difference is tiny and experimentally insignificant. The set of proteins that is the highlight of the paper, shown in Figure 4C and referred to in the last sentence of the abstract (... the largest set of orthogonal PPIs to date...), has an orthogonality gap of just 0.01. I asked about this value in my review, and the authors provided it in their rebuttal. It is also available in the manuscript, although it is not emphasized or easy to find. The authors state the estimated error of the assay as ~0.37 interaction score units, which I think is overly optimistic (see below). But even this estimate makes a difference of 0.01 negligible.

In order to make our results more useful to the scientific community we have chosen a more conservative cutoff of 1.0 Interaction Score unit for the sets displayed in the main figures. With the stricter cutoff we obtain a set of 15 orthogonal pairs (shown in figure 4C) instead of 22 with a 0.01 cutoff. The 15 pairs set has an orthogonality gap of 1.27 and is twice the size of the current state of the art coiled coil set (7 pairs by Mason et. al.) and larger than the 12 pair set developed by Chen et. al.

The even larger orthogonal sets with narrower orthogonality gaps are still available in tables S6 and S7.

The authors have a considerable amount of other data that point to larger uncertainty, particularly the error that they get when reversing the fusion pairs (swapping proteins “1” and “2” – in Figure 1E - or “X” and “Y” – Figure 1H and S17, where no RMS error is reported, only a correlation coefficient) and the error when they re-run the CC0 library repeatedly (Figure S18; other analyses could also be done using the CC0 replicates to add rigor). There are many tests that the authors could do, using the data they already have, to estimate the error more appropriately and then apply this as a reasonable

cutoff for an orthogonality gap. Even in the absence of further experimental validation, this could provide some reasonable support for new orthogonal sets. They don't do this, however, and as a result, the labeling of many of the sets presented here as "orthogonal" is not justified. Although the processed data are available as part of the paper, and researchers using these data can use whatever orthogonality cutoff they prefer, the use of unjustified cutoffs to define orthogonal sets that are a major claim of the paper is not appropriate.

We have performed the detailed analysis suggested by the reviewer. The results are summarized in table S9. In addition to looking at the standard deviation of the difference between two technical repeats (data from Fig S18, first row of table S9), we have also analyzed the asymmetry, that is the standard deviation of the difference of the interaction score between the two orientations of the split enzyme (i.e. $\text{StdDev}(XY-YX)$). Since we have noticed that there were a few random high signal outliers, we have also computed the median absolute deviation (MAD) which is less susceptible to outliers. However in the asymmetry of the CCmax library the StDev and MAD values were very similar so we chose the more common StDev.

The standard deviation for the entire CCMax library is 0.84. We have rounded that up to 1 and used that as our orthogonality gap. The code for the analysis is available at https://github.com/ajasja/NGB2H/tree/main/07_supplementary_files/uncertainty_estimation.

(“To make our results robust to experimental noise from the NGB2H assay, we needed to find an appropriate orthogonality gap, that is larger than the uncertainty of the interactions score. We have performed a thorough analysis of both the CC0 internal control (technical repeats), external controls (comparison to measured melting points, Figure S10C) and especially the availability of reciprocal enzyme orientations for the same peptide pair (pairs of identical peptides where the split cAMP parts are reversed). We found uncertainty of less than 0.8 interaction scores in all experiments in this paper (Table S9). Thus, to be conservative we enforced an orthogonality gap of at least 1.0 Interaction Score. Using this framework, we were able to identify an orthogonal subset of coiled-coils that contains six pairs, which includes one heterodimer and five homodimers (Figure 2B). The orthogonality gap we enforce is very strict, for example the CC0 control set has a gap of only 0.4, and at the orthogonality gap of 1.0 it contains only four pairs instead of seven.

There are also applications where the requirements for orthogonality can be reduced, for example in building protein origami as demonstrated by Aupič et al.²¹, in which two identical pairs were used in the same structure. Pairwise orthogonality is the most stringent criterion. In a single-pot experiment, in which all pairs would be present, we speculate that orthogonality would only improve because the off-target states would be competing with the on-target states.

Therefore we have also calculated orthogonal sets with orthogonality gaps of 0.0 and 0.5. At an orthogonality gap of zero, 20 of our 51 experimentally identified orthogonal subsets in CCNG1 library had more than the seven on-target orthogonal interactions (Figure S13). Orthogonal sets at different orthogonality gaps are presented in Table S6.”)

When looking into the reproducibility, I noticed that sometimes the interaction matrices are reported as full matrices (Figures 1H, 4C, S20) and sometimes as half matrices (Figures 2B and C). In Figure 1H, there is noise when comparing the reciprocal interactions, as expected. Figure S17E suggests there should be quite a lot of noise. But to the eye, Figure 4C appears perfectly symmetrical. This is highly implausible. If the matrix was symmetrized, then it is very misleading to show the whole thing. The experimental asymmetry provides a highly relevant measure of expected uncertainty that would be appropriate to use in the analysis, but it seems not to be included in Figure 4C. It looks like the matrices in Figure S20 have also been symmetrized.

The matrices were indeed symmetrized, since the symmetric matrices were used to find orthogonal sets. We agree that asymmetry would be valuable and so we have put asymmetric versions in all the figures (Fig 1H, Fig 4C, S8, S9, S20). Where we do not have asymmetry information available (in library CCNG1), we render only half of the matrix (Fig 2B).

I agree with the authors that providing these data and models to the community will make a meaningful contribution. There are many interesting methods and results in this work. I also recognize that in their current circumstances, they can't do more experiments. However, in its current form, I think the manuscript is misleading and makes claims about orthogonality that are not supported by the data. Also, I will again raise Figure 2E, which I continue to find unconvincing despite the authors' rebuttal. A fit to a polynomial that is statistically better than a fit to a line, when both fits are extremely poor and the trend is not well-defined by available data, does not provide good support for their claim.

In order to avoid confusion, we have removed Fig 2E.

Finally, although not a criterion for acceptance (since the sequences are all given) it seems that the authors are not making the clones available (e.g., in AddGene), which is regrettable.

We thank the reviewer for the good suggestion! We are in the process of depositing the plasmids with Addgene. We have left placeholders for the add gene IDs in the text, which can be added during the proof stage.

("Plasmids psk33, 34, 59, 168 and 179 are available in the Addgene repository (<https://www.addgene.org/>) under accession codes AAAAAA, BBBBBB, CCCCCC, DDDDDD, EEEEE and FFFFFF")